# EfficientLLM: Unified Pruning-Aware Pretraining for Auto-Designed Edge Language Models

## Abstract

Modern large language models (LLMs) driven by scaling laws achieve emergent intelligence in large model sizes. Recently, the increasing concerns about cloud costs, latency and privacy make it an urgent requirement to develop compact edge language models. Distinguished from direct pretraining that bounded by the scaling law, this work proposes the pruning-aware pretraining, focusing on retaining performance of much larger optimized models. It features following characteristics: 1) Data-Scalable Pruning: we introduce minimal parameter groups in LLM and continuously optimize structural pruning, extending post-training pruning methods like LLM-Pruner and SparseGPT into the pretraining phase. 2) Auto-Designed Architecture: the LLM architecture is auto-designed using saliency-driven pruning, which is the first time to exceed SoTA human-designed LLMs in modern pretraining. We reveal that it achieves top-quality edge language models, termed EfficientLLM, by scaling up LLM compression and extending its boundary. EfficientLLM significantly outperforms SoTA baselines with $100M \sim 1B$ parameters, such as MobileLLM, SmolLM, Qwen2.5-0.5B, OLMo-1B, Llama3.2-1B in commen sense benchmarks. As the first attempt, EfficientLLM bridges the performance gap between traditional LLM compression and direct pretraining methods, and we fully open source EfficientLLM for future advancements.

## 1 Introduction

Large Language Models (LLMs) have become a central component of modern AI systems (Achiam et al., 2023; Guo et al., 2025) and are increasingly transforming daily life, particularly in mobile edge applications. However, typical LLMs (Touvron et al., 2023a), with 7 billion to 1 trillion parameters, require on-cloud deployment and continuous internet connectivity for interface. This places significant challenges in terms of latency, data-security and cloud-costs. In fact, fully relying on LLMs for mobile edge applications can be impractical — serving all mobile applications with GPT-4 (Achiam et al., 2023) would require approximately one million H100 GPUs (Liu et al., 2024b). As a result, developing edge language models on resource-constrained devices becomes a recent tendency. For instance, MobileLLM (Liu et al., 2024b) focuses on sub-one billion model sizes, which would fit in the DRAM of smartphones without excessive consumption.

Direct pretraining is dominant in recent tiny language model pretraining. Some practices such as MobileLLM and PanGu-$\pi$-Pro (Tang et al., 2024) design deep-and-thin architectures for model efficiency. Other practices such as TinyLlama (Zhang et al., 2024a) and Qwen2.5-0.5B (Yang et al., 2024b) focus on scaling up pretraining data to 3T and 17T tokens. Based on best architectures and sufficient data, modern tiny models (Yang et al., 2024b; Groeneveld et al., 2024) are showing promise in reaching performance boundary. However, their overall performance appears to be somewhat locked by the parameter scaling law (Kaplan et al., 2020): given limited model size, simply scaling up pretraining data is inefficient. More importantly, the emergent intelligence (Brown et al., 2020) is only observed on larger model sizes, meaning tiny models may never acheve this by direct pretraining alone. What is the next to train more efficient edge models remains an open challenge.

In parallel, LLM compression (Ashkboos et al., 2024; Gu et al., 2024; Han et al., 2015) focus on retaining the performance of larger and stronger models while reducing computational cost. Despite

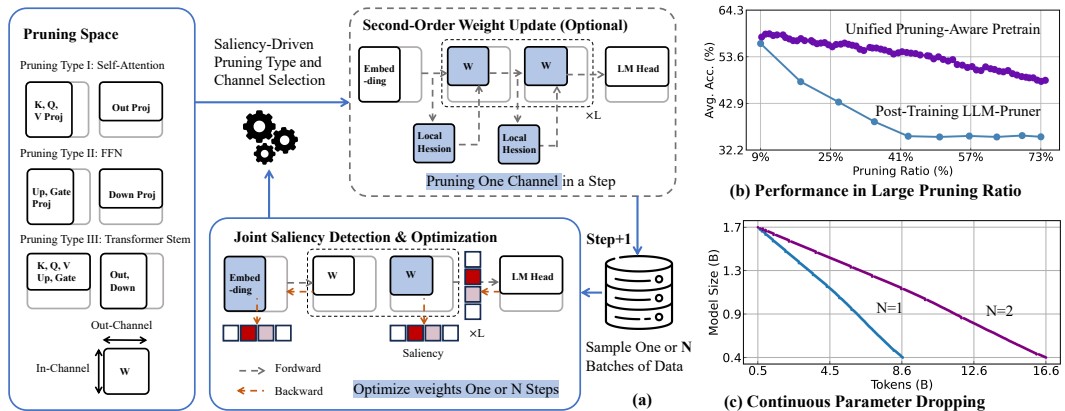

Figure 1: An overview of unified pruning-aware pretraining. (a) Towards **unified pruning, pretraining, and architecture design**, the training loop includes: joint saliency detection and weight optimizing, pruning type selection from pruning space, and weight updating. (b) Improve post-training pruning by pruning-aware pretraining. (c) Continuous model compression during pretraining.

its potential efficiency, existing methods (Sreenivas et al., 2024; Frantar & Alistarh, 2023; Xiao et al., 2023) compress LLM only using a small calibration dataset in post-training, which often results in significant performance degradation, making them unsuitable for top-quality edge language models. Recently, ShearedLlama (Xia et al., 2023) initializes from an optimized LLM, improving training efficiency. However, the constrained optimization (Platt & Barr, 1987) hinders scaling up pruning stage and the performance gap to direct pretraining still remains. This work extends the performance boundary of traditional LLM compression by scaling up training data, a crucial but underexplored approach in the LLM compression field.

This work proposes the unified pruning-aware pretraining to extend the efficiency boundary of edge language models. A family of top-efficiency edge language models in $100M \sim 1B$ sizes are pretrained, named EfficientLLM. As shown in Fig. 1, we formulate pruning-aware pretraining as a unified framework for weight pruning, pretraining, and architecture design: 1) Compared with direct pretraining, pruning-aware pretraining leverages the performance of much larger optimized models, which direct pretraining smaller models never achieves. 2) Compared with post-training pruning, it scales up the pruning stage with pretraining data. As shown in Fig. 1 (b), pruning-aware pretraining scales up vanilla LLM-Pruner, achieving more than a 10% increase in accuracy. 3) Driven by saliency, the overall architecture can be auto-designed (Yu et al., 2020; Zoph et al., 2018) according to a predefined pruning space step by step.

This work advances both edge language models and LLM compression:

- We propose a family of SoTA edge language models in $100M \sim 1B$ sizes, named EfficientLLM. EfficientLLM significantly exceeds direct pretrained tiny models by unified and scalable pruning.

- We propose the unified pruning-aware pretraining, promoting LLM compression to the era of pretraining. General post-training methods like LLM-Pruner (Ma et al., 2023), SparseGPT (Frantar & Alistarh, 2023), and Wanda (Sun et al., 2023) could be embedded. By scaling up the pruning stage, vanilla LLM-Pruner significantly exceeds SoTA methods without bells and whistles.

- We explore the auto-designed architectures in modern pretraining for the first time. Saliency-driven architectures are auto-searched via unified pruning and competitive with human practices.

## 2 PRELIMINARY AND RELATED WORKS

**Edge Language Models.** Modern large language models follow the scaling law (Kaplan et al., 2020): larger models achieve higher data efficiency, making optimal training favor large models with moderate data. Towards accurate compact models, a lot of efforts explore the optimal training recipes: 1) data scale. OLMo-1B (Groeneveld et al., 2024), TinyLlama-1.1B (Zhang et al., 2024a), Qwen2.5-0.5B (Yang et al., 2024b) pretrain on 2T, 3T, and 17T tokens respectively, which is significantly larger than the optimal data sizes according to scaling law. 2) Architectures. MobileLLM (Liu et al.,

2024b) shows that the deep-and-thin network and layer sharing achieve additional performance gains. However, direct pretraining is bounded by the scaling law, and can be data-inefficient. More recently, Llama3.2 (Dubey et al., 2024) and MiniTron (Sreenivas et al., 2024) introduce distillation and pruning for data-efficient training. There are mainly 2 drawbacks which addressed in this work: 1) The LLM pruning itself does not scale up. MiniTron only uses a small calibration dataset for pruning and only scales up recovery training, while this work scales up pruning itself to retain more performance. 2) Knowledge distillation (Gu et al., 2024; Ko et al., 2024) during pretraining is not training-efficient, as teacher models are typically 7B-scale LLMs (Touvron et al., 2023b), consuming $7 \sim 50\times$ FLOPs than sub-billion edge models, which we avoid in EfficientLLM. For deployments, quantization (Shao et al., 2023; Xiao et al., 2023; Liu et al., 2024a) can also be adopted.

**LLM Pruning** (Dong et al., 2024; Zhang et al., 2024b; Zhao et al., 2024; Bhaskar et al., 2024). We mainly focus on structural pruning to address hardware friendly edge language models. The most widely used LLM pruning is based on the Taylor expansion (LeCun et al., 1989; Hassibi et al., 1993; van der Ouderaa et al., 2023). By calibration, typical SparseGPT (Frantar & Alistarh, 2023) and Wanda (Sun et al., 2023) can only applied in semi-structured pruning; LLM-Pruner (Ma et al., 2023) only achieves 20% pruning ratio with reasonable accuracy. Even if pruning with finetuning, LoraPrune (Zhang et al., 2023a) can only prune in 50% ratio. So there is an urgent requirement to scale up LLM pruning in pretraining. Another line of works learn to initialize from source model such as ShearedLlama (Xia et al., 2023) and NutePrune (Li et al., 2024) with less than 0.5B tokens. However, ShearedLlama needs human-designed target and does not explore scalability in large scale data; and this work explore unified pruning-aware pretraining with Taylor expansion.

## 3 UNIFIED PRUNING-AWARE PRETRAINING

According to scaling laws, both the scale of training data and the number of parameters are fundamental to the emergence of intelligence in modern LLMs. Direct pretraining of smaller models is inefficient and lacks generalization ability. Model compression methods, although based on pretrained large models, fail to meet the data scale requirements and suffer from significant performance drop.

The principle of this work is to bridge the gap between direct pretraining and LLM compression by a unified training scheme. In practice, pruning-aware pretraining continuously drops parameters in training, which integrates pruning, pretraining, and architecture design at the same time.

**Problem Formulation.** Finding a sub-network from a pretrained LLM is non-trivial. Given an optimized LLM, post-training LLM pruning focuses on finding optimal channels in each layer towards a target architecture. However, for edge language models, it is still challenging to define the efficient target architecture from its source model. For instance, MobileLLM shows the deeper architecture is better than the wider for sub-billion LLMs by human design and practice. However, this best practice may be sub-optimal for a given source model, as each model may exhibit unique saliency patterns that suggest different pruning targets. We formulate the architecture-agnostic pruning problem as:

$$\min_{a \in \mathcal{A}} \min_{c \in \mathcal{C}} \min_{w} \mathcal{L}_{pretrain}(a, c, w | \mathcal{M}), \tag{1}$$

where $\mathcal{A}$ and $\mathcal{C}$ are sub-architectures (Wu et al., 2019; Liu et al., 2018) and sub-channels sampled from the source model $\mathcal{M}$. We jointly optimize pretraining loss through three factors: 1) the sub-architecture, $a$, 2) the sub-channels, $c$, and 3) the model weights, $w$. We outline the pruning-aware pretraining in Fig. 1 and detail each part in the following subsections.

### 3.1 DEFINING MINIMAL PRUNING GROUP

To design architectures automatically via pruning, we first define the minimal parameter groups as the minimal unit to prune in each step, which should be flexible enough to construct any shape transformers after pruning. Given a pretrained source model $\mathcal{M}$, the pruned model $\mathcal{M}^*$ can be:

$$\mathcal{M}^* = \mathcal{M} - \sum_{t=1}^{n} g_t, \quad \text{s.t.} \quad \min_{g_t \in \mathcal{G}} \mathcal{L}_{pretrain}(\mathcal{M}), \tag{2}$$

where $g_t$ is the mini-group of parameters pruned in step t, and $\mathcal{G}$ is the pruning space formulated by defined mini-groups. According to Eq. 2, the pruning can be approximately decoupled by t steps and

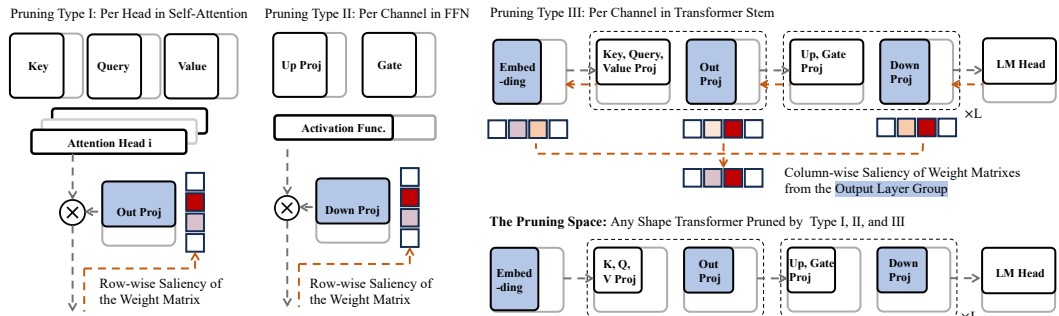

Figure 2: Three basic pruning types in the pruning space. We plot all the weight metrics with shape $[D_{input}, D_{output}]$. In backpropagation (in orange), the saliency of the output layer group (in blue) is calculated according to Eq. 9.

solved sequentially:

$$\mathcal{M}_t = \mathcal{M}_{t-1} - g_t^* \quad \text{s.t.} \quad g_t^* = \underset{g_t \in \mathcal{G}}{\operatorname{argmin}} \; \mathcal{L}_{pretrain}(g_t | \mathcal{M}_{t-1}). \tag{3}$$

We first assume an optimal $g_t^*$ in each pruning step to minimize pretraining loss, and solve how to acquire $g_t^*$ in the next subsection. In each pruning step, an optimal mini-group of parameters are selected and dropped from the pretraining LLM $\mathcal{M}_{t-1}$, allowing the source model $\mathcal{M}$ to adaptively reduce the number of parameters until a specific computation budget is met.

For the fully structured pruning space, we impose two constraints in the design of mini-groups: 1) the hidden size, attention heads and intermediate size can be pruned flexibly; 2) the shape of different layers is the same. Unlike LLM-Pruner pruning space, which relies on manually specified pruning targets, our approach divides the pruned parameters into different types of minimal groups that can be adaptively combined during pretraining.

**Parameter Mini-Groups.** For simplify, we indicate the ***input layer group*** as the *query, key, value projections* in attention blocks; or the *up, gate projections* in feed forward blocks. We indicate the ***output layer group*** as the *output projections* in attention blocks; or the *down projections* in feed forward blocks. As shown in Fig. 2, we define three basic pruning types and their mini-groups, $\mathcal{G}_{attn}$, $\mathcal{G}_{ffn}$, $\mathcal{G}_{stem}$:

1) Per-head pruning in self-attention blocks: when an attention head is pruned, all the corresponding output channels in the input layer group and input channels in the output layer group are pruned at the same time. We select the mini-group $\mathcal{G}_{attn}^{(\ell)}$ with the minimal saliency in the $\ell_{th}$ layer, and merge $\mathcal{G}_{attn}^{(\ell)}$ in all layers as $\mathcal{G}_{attn}$:

$$\mathcal{G}_{attn} = \{W_{:,i:j}^{(k,\ell)}, W_{:,i:j}^{(q,\ell)}, W_{:,i:j}^{(v,\ell)}, W_{i:j,:}^{(o,\ell)}, \ell = 1, 2, ..., n\}, \tag{4}$$

where $W_{:,i:j}$ and $W_{i:j,:}$ are column-wise and row-wise pruned; i:j corresponds to channels of an attention head; and n is the overall blocks.

2) Per-channel pruning in feed-forward blocks: when a intermediate channel is pruned, the coupled channels include one output channel in the input layer group, and one input channel in the output layer group in. We couple the minimal-saliency group $\mathcal{G}_{ffn}^{(\ell)}$ in the $\ell_{th}$ layer and merge $\mathcal{G}_{ffn}^{(\ell)}$ in all layers as $\mathcal{G}_{ffn}$:

$$\mathcal{G}_{ffn} = \{W_{:,i}^{(up,\ell)}, W_{:,i}^{(q,\ell)}, W_{i,:}^{(down,\ell)}, \ell = 1, 2, ..., n\}. \tag{5}$$

3) Per-channel pruning in the transformer stem: When a channel of the transformer stem is pruned, one channel in the token embedding, one input channel in input layer group and one output channel in output layer group for every block, one input channel of the LM head projection is correspondingly pruned at the same time. We donate the stem mini-group as $\mathcal{G}_{stem}$:

$$\mathcal{G}_{stem} = \{W_{i,:}^{(k,\ell)}, W_{i,:}^{(q,\ell)}, W_{i,:}^{(v,\ell)}, W_{:,i}^{(o,\ell)}\}, ... \tag{6}$$
$$\cup \{W_{i,:}^{(up,\ell)}, W_{i,:}^{(gate,\ell)}, W_{:,i}^{(down,\ell)}\}, ...$$
$$\cup \{\mathbf{w}_i^{(emb)}, W_{i,:}^{(head)}\}, \quad \ell = 1, 2, ..., n$$

where i should be the same in every blocks in Eq. 6; i,j needn't the same across blocks in Eq. 4,5.

Given a transformer with hidden size $m$, head number $h$, intermediate size $n$, and $l$ layers, the original pruning space is $h^\ell \times n^\ell \times m$. In each pruning step, the mini-groups are dynamically grouped by saliency, and we only choose among the 3 types to prune. By coupling the parameters into mini-groups, the choice space is reduced to 3 in each step of Eq.3, and the final pruning space is $3^t$.

### 3.2 OPTIMIZING MINI-GROUPS BY SALIENCY

Based on the mini-groups, Eq.1 becomes a bi-level optimization problem of the mini-groups $g$ and weights $w$:

$$\min_{g \in \mathcal{G}} \mathcal{L}_{pretrain}(g, w^*|\mathcal{M}), \quad \text{s.t.} \quad w^* = \operatorname*{argmin}_w \mathcal{L}_{pretrain}(w, g^*|\mathcal{M}), \tag{7}$$

where the outer optimization could be solved by pruning a mini-group in each step as Eq.3, and the inner optimization could be directly solved by weight pretraining. The weight pretraining step and mini-group optimization (Eq.3) step alternate, and the model size drops continuously during pretraining as shown in Fig. 1 (c), until the pre-defined parameter budget is achieved.

*Different from vanilla iterative pruning along a predefined trajectory, unified pruning selects its pruning trajectory based on saliency.* The pruned model is automatically optimized toward the most salient sub-architectures. Thanks to large-scale pretraining data, we find that these saliency-driven architectures are competitive with human-designed ones, effectively eliminating the need for manual pruning target design and repeated trial-and-error.

**Mini-Group Saliency.** In each mini-group selection step, Taylor expansion evaluates the optimal mini-group $g_t^*$ in Eq.3. For an optimized source model, loss of any weight $\mathbf{w}$ can be approximated by a second-order Taylor expansion around its optimal value $\mathbf{w}^*$:

$$\mathcal{L}(\mathbf{w}) \simeq \mathcal{L}(\mathbf{w}^*) + \delta\mathbf{w}^\top \nabla\mathcal{L}(\mathbf{w}^*) + \frac{1}{2}\delta\mathbf{w}^\top \mathbf{H}_\mathcal{L}(\mathbf{w}^*)\delta\mathbf{w} \tag{8}$$

where $\mathcal{L}$, $\nabla\mathcal{L}$, $\mathbf{H}_\mathcal{L}$ is the global loss, gradient, hessian matrix; and $\delta\mathbf{w} = \mathbf{w} - \mathbf{w}^*$. We substitute Eq.8 into Eq.3:

$$g_t^* = \operatorname*{argmin}_{g_t \in \mathcal{G}} \mathcal{L}_{pretrain}(g_t|\mathcal{M}_{t-1}) \tag{9}$$

$$= \operatorname*{argmin}_{g_t \in \mathcal{G}} g_t^\top \nabla\mathcal{L}(\mathcal{M}_{t-1}) + \frac{1}{2}g_t^\top \mathbf{H}_\mathcal{L}(\mathcal{M}_{t-1})g_t,$$

where we omit the first term $\mathcal{L}(\mathbf{w}^*) = \mathcal{L}(\mathcal{M}_{t-1})$ in Eq.8, because $\mathcal{L}(\mathcal{M}_{t-1})$ is the same in the step t for the 3 mini-groups, $\mathcal{G} = \{\mathcal{G}_{\text{attn}}, \mathcal{G}_{\text{ffn}}, \mathcal{G}_{\text{stem}}\}$. And we could calculate mini-group saliency according to Eq. 9 (Ma et al., 2023).

**Efficient Calculation.** In practice, we only calculate the saliency of the output layer groups for efficiency. a neural network is a directed acyclic graph (DAG) (Liu et al., 2018). For each node in the graph, pruning all its inputs or all of its outputs is sufficient to prune the entire network. It saves $2 \sim 3$ times computation with output layer group only calculation. Details are shown in Fig. 2: 1) *Pruning Type I*: we only calculate element-wise saliency matrix for the weights of the output projection, and then sum each column of the saliency matrix. We select $\mathcal{G}_{\text{attn}}^{(\ell)}$ based on the lowest row-wise saliency in the output projection weights. 2) *Pruning Type II*: we only calculate the element-wise saliency matrix for the down projection, and then sum each column. $\mathcal{G}_{\text{ffn}}^{(\ell)}$ with the lowest row-wise saliency are selected. 3) *Pruning Type III*: we already have all the element-wise saliency in the output layer group based on Type I and II. To evaluate saliency of the hidden state, we first sum each row of saliency matrices, and then, sum the saliency in all of the output layer group.

**Hessian Approximations.** Existing post-training methods such as LLM-Pruner (Ma et al., 2023), SparseGPT (Frantar & Alistarh, 2023), and Wanda (Sun et al., 2023) have proposed various Hessian approximations to approximate hessian matrices. By substituting Eq. 9, our framework can naturally extend these post-training pruning methods to the unified pretraining stage. Without loss of generality, we also generalize the second-order weight updating to pretraining in the next subsection.

### 3.3 SECOND-ORDER WEIGHT UPDATING

Existing second order pruning applies the same Hessian matrix for the pruning weight detection and the remaining weight updating. However, calculating the global Hessian matrix is impossible in modern LLMs for its $\mathcal{O}(n^4)$ complexity. A common approach is to use the squared error at each layer as a proxy for the global loss: $\mathbf{H}_{\mathcal{L}} \simeq XX^T$, such as in SparseGPT (Frantar & Alistarh, 2023), OBC (Frantar & Alistarh, 2022). Although achieving the $\mathcal{O}(d_{row} \times d_{col}^2)$ complexity, Hessian matrices can not describe the global loss.

This work addresses this problem by decoupling the Hessian matrix in saliency detection and weight updating. To capture global saliency, we approximate with global diagonal Hessian matrices as LLM-Pruner for saliency detection; to reduce computational complexity, we apply the layerwise proxy Hessian, $\mathbf{H}_{\mathcal{L}} \simeq XX^T$, for weight updating. In each step, we prune a mini-group including only one column of weights in a layer, and the remaining weights are updated by $\delta w_p = -\frac{w_p}{[\mathbf{H}^{-1}]_{pp}} \cdot \mathbf{H}_{:,p}^{-1}$.

To efficiently compute the $p$-th column of the inverse Hessian matrix $\mathbf{H}_{:,p}^{-1}$, it suffices to solve the linear equation $\mathbf{H}\mathbf{H}_{:,p}^{-1} = \mathbf{e}_p$ in a weight updating step.

## 4 EXPERIMENTS

**Models.** To compare with the most general post-training pruning, EfficientLLM-A basically approximates Eq.9 as LLM-Pruner. EfficientLLM-B additionally applies the second-order weight updating based on EfficientLLM-A. 1) In empirical studies, we evaluate EfficientLLM with SmolLM-1.7B (Allal et al., 2024), Llama2-7B (Touvron et al., 2023b), and Qwen2.5-7B (Yang et al., 2024b) as the source models. 2) In main results, we prolong and pretrain EfficientLLM-134M from the source model SmolLM-360M; EfficientLLM-457M and 1.1B from SmolLM-1.7B. 3) In comparisons with LLM pruning, we keep the Llama-7B (Touvron et al., 2023a) source model.

**Data Composition.** EfficientLLM maintains a data distribution similar to the source model: 1) in main results, our pretraining data composition is similar to SmolLM, including 220B tokens from FineWeb-Edu (Lozhkov et al., 2024), 28B tokens from Cosmopedia v2 (Ben Allal et al., 2024a), 4B tokens from Python-Edu (Ben Allal et al., 2024b), and 27.5B tokens randomly sampled from OpenWebMath (Paster et al., 2023). 2) In comparisons with LLM pruning, we sample from RedPajama-1T (Weber et al., 2024) as pretraining data with the Llama family as the source model.

**Training.** In main results, we use the 2 stage pretraining as ShearedLlama (Xia et al., 2023) but explore in large scale: the large scale unified pruning followed by the continued pretraining. For EfficientLLM-134M, 460M, and 1.1B, we pretrain 50.3B, 72.1B, and 36.7B tokens for unified pruning followed by 500B, 500B, and 320B tokens continued pretraining. Note that the large-scale continued pretraining is not necessary, and 50B tokens also achieve competitive performance. Note also that the number of tokens used for unified pruning is determined by the number of iterations required to reach the target parameter count. All the training details are shown in Appendix B.1.

**Evaluations.** For pretrained base models, we follow Llama, MobileLLM, and ShearedLlama to evaluate Common Sense Reasoning tasks: ARC (Clark et al., 2018), BoolQ (Clark et al., 2019), HellaSwag (Zellers et al., 2019), OBQA (Mihaylov et al., 2018), PIQA (Bisk et al., 2020), and WinoGrande (Sakaguchi et al., 2021). The MMLU (Hendrycks et al., 2020) for Word Knowledge evaluation is also applied. For instruct finetuned model, we use the standard Alpaca-Eval (Li et al., 2023) with GPT-4o as the judge model.

### 4.1 EMPIRICAL STUDIES

As shown in Table 1, 2, 3, we train around 450M target models from the source model SmolLM-1.7B.

**Training Schemes.** We first compare the efficiency of different training schemes under the same 5B token budget: 1) Direct Pretraining: three architectures are pretrained from scratch using 5B tokens: (i) Direct-hidden: Prunes the hidden size of the source architecture. (ii) Direct-source: Uniformly scales down the source model. (iii) Searched architecture: Uses the architecture searched by Unified Pruning. 2) ShearedLlama: Requires manually specified pruning targets. We consider two variants: pruning hidden size or uniformly scaling the source model. 3) Unified Few-shot Pruning: Applies

Table 1: Comparison of different training schemes under **the same token budget**. 1) Direct pretraining: "hidden" or "source" indicates target architectures from the source model. 2) Comparison with ShearedLlama in >70% pruning ratio. 3) Comparison with LLM-Pruner with continued pretraining in smaller pruning ratios. We prune both LlaMA2-7B and Qwen2.5-7B for 4000 steps to 3.3B and 5.2B.

| Model | Source | ARC-c | ARC-e | BoolQ | HellaSwag | OBQA | PIQA | WinoGrande | Avg. |
|---|---|---|---|---|---|---|---|---|---|
| Direct-hidden | – | 26.88 | 55.85 | 52.39 | 37.10 | 30.20 | 66.65 | 50.51 | 45.65 |
| Direct-source | – | 28.67 | 59.68 | 56.30 | 38.41 | 32.00 | 65.67 | 49.96 | 47.24 |
| Searched Arch. (Ours) | – | 29.10 | 59.30 | 61.38 | 37.74 | 32.80 | 66.76 | 51.07 | 48.30 |
| ShearedLlama-hidden | SmolLM-1.7B | 28.41 | 57.41 | 60.98 | 39.79 | 30.80 | 66.81 | 54.06 | 48.32 |
| ShearedLlama-source | SmolLM-1.7B | 30.89 | 62.08 | 61.07 | 44.29 | 32.60 | 68.39 | 52.25 | 50.22 |
| Unified FewShot Prun. | SmolLM-1.7B | 30.63 | 62.67 | 61.22 | 44.31 | 34.00 | 68.66 | 53.43 | 50.25 |
| **EfficientLLM-A** | SmolLM-1.7B | 30.46 | **64.06** | **61.99** | 45.98 | 34.00 | **69.91** | 53.91 | 51.47 |
| **EfficientLLM-B** | SmolLM-1.7B | **30.97** | 63.22 | 60.86 | **46.51** | **35.00** | 69.64 | **55.09** | **51.61** |
| LLM-Pruner | LLaMA2-7B | 27.30 | 43.52 | 61.10 | 38.15 | 29.40 | 63.11 | 51.14 | 44.82 |
| **EfficientLLM-A** | LLaMA2-7B | **32.76** | **57.53** | **65.96** | **53.07** | **33.80** | **69.42** | **58.09** | **52.95** |
| LLM-Pruner | Qwen2.5-7B | 38.40 | 66.67 | 70.58 | 57.97 | 38.40 | 72.52 | 60.62 | 57.88 |
| **EfficientLLM-A** | Qwen2.5-7B | **40.10** | **70.50** | **76.79** | **61.26** | **40.80** | **73.34** | **61.72** | **60.64** |

Table 2: Generalization to various pruning metrics. LLM-Pruner (Ma et al., 2023), OBC (Frantar & Alistarh, 2022), Diag-Hessian (Sun et al., 2023; LeCun et al., 1989) metrics are embedded in the unified pruning. All results are without continued pretraining.

| Model | ARC-c | ARC-e | BoolQ | HellaSwag | OBQA | PIQA | WinoGrande | Avg. |
|---|---|---|---|---|---|---|---|---|
| LLM-Pruner | 22.18 | 26.64 | 49.63 | 25.62 | 27.80 | 51.20 | 50.59 | 36.24 |
| +Unified Pruning | 29.18 | 57.79 | 59.57 | 41.93 | 34.40 | 66.97 | 52.88 | **48.96** |
| OBC (SparseGPT) | 25.51 | 25.88 | 37.86 | 26.81 | 29.20 | 51.41 | 50.51 | 35.31 |
| +Unified Pruning | 29.44 | 60.02 | 62.02 | 42.27 | 33.40 | 67.41 | 53.35 | **49.70** |
| Diag-Hess (Wanda, OBD) | 25.85 | 25.88 | 50.80 | 26.06 | 30.40 | 51.69 | 48.22 | 36.99 |
| +Unified Pruning | 29.95 | 60.14 | 60.83 | 41.68 | 32.80 | 66.38 | 52.88 | **49.24** |

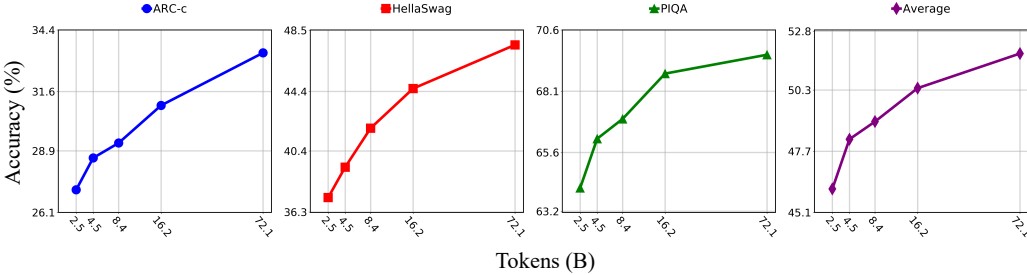

Figure 3: Scalability of unified pruning-aware pretraining without continued pretraining.

Table 3: Ablation studies on the unified pruning-aware pretraining. "Direct Training" keeps the same architecture ratio as the source model. "Pruned Arch." indicates the auto-designed architecture.

| Model | ARC-c | ARC-e | BoolQ | HellaSwag | OBQA | PIQA | WinoGrande | Avg. |
|---|---|---|---|---|---|---|---|---|
| Direct Training | 28.67 | 59.68 | 56.30 | 38.41 | 32.00 | 65.67 | 49.96 | 47.24 |
| +Searched Arch. | 29.10 | 59.30 | 61.38 | 37.74 | 32.80 | 66.76 | 51.07 | 48.30 |
| +Unified FewShot Pruning | 30.63 | 62.67 | 61.22 | 44.31 | 34.00 | 68.66 | 53.43 | 50.25 |
| +Prolong Pruning | 34.13 | 66.16 | 60.49 | 49.87 | **35.80** | 70.67 | 54.30 | 53.06 |
| +Prolong Continued Pretrain. | **35.92** | **70.50** | 59.85 | **53.16** | 35.00 | **72.69** | **56.27** | **54.77** |

mini-group optimization on 128 samples using LLM-Pruner's metric, followed by training on 5B tokens. 4) Unified Pruning-aware Pretraining: Follows the ShearedLlama pipeline, pruning with 1B tokens, then continuing training on the remaining 4B tokens.

We evaluate the necessity of unified architecture auto-design, pruning, and pretraining in Table 1: 1) **Architecture matters**: For directly pretrained edge models, the choice of architecture significantly impacts performance. The searched Arch. outperforms Direct-hidden by 2.7%. 2) **Pruning boosts edge models**: With the same architecture, Unified Pruning-aware Pretraining surpasses Search Arch. by 3.17%. 3) **Scaling up pruning can benefit more than finetuning**: Compared to Unified Fewshot Pruning, Pruning-aware pretraining allocates more tokens to the pruning stage and improves accuracy by 1.22%. Unlike ShearedLlama, Unified Pruning requires no manual architecture design. For example, ShearedLlama-hidden adopts a suboptimal target, resulting in 3.15% accuracy drop.

**Pruning Metrics.** We embed well-studied post-training pruning metrics in our Unified Pruning-aware Pretraining, enhancing performance of existing methods. By replacing Eq. 9, we apply LLM-Pruner

Table 4: Zero-shot performance on World Knowledge and Common Sense Reasoning tasks. "Avg." calculate among the 7 Common Sense Reasoning tasks. #Tokens count continued pretraining for EfficientLLM. All the results are evaluated on the same evaluation (Appendix B.2).

| Model | #Tokens | #Params | MMLU | ARC-c | ARC-e | BoolQ | HellaSwag | OBQA | PIQA | WinoGrande | Avg. |
|---|---|---|---|---|---|---|---|---|---|---|---|
| OPT-125M | 180B | 125M | 26.02 | 22.87 | 43.31 | 55.44 | 31.37 | 27.80 | 62.62 | 49.80 | 41.89 |
| GPT-neo-125M | 300B | 125M | 26.89 | 23.29 | 43.22 | 61.77 | 30.49 | 26.00 | 62.62 | 51.93 | 42.76 |
| Pythia-160M | 300B | 162M | 26.43 | 22.27 | 37.84 | 43.33 | 29.97 | 26.40 | 58.87 | 49.96 | 38.38 |
| Mamba-130M | 1.2T | 130M | 27.65 | 24.49 | 47.56 | 54.68 | 35.11 | 29.00 | 64.69 | 53.35 | 44.13 |
| MobileLLM-125M | 1T | 125M | 27.58 | 24.32 | 46.38 | 60.34 | 38.15 | 28.40 | 65.13 | 52.41 | 45.02 |
| SmolLM-135M | 600B | 135M | 30.05 | 29.35 | 61.32 | 59.85 | 42.67 | 34.40 | 68.55 | 52.96 | 49.87 |
| EfficientLLM-A | 500B | 134M | 30.54 | 30.97 | 62.88 | 60.40 | 43.81 | 33.60 | 68.82 | 53.28 | 50.54 |
| OPT-350M | 180B | 331M | 26.96 | 23.98 | 44.02 | 57.80 | 36.63 | 27.80 | 64.91 | 52.96 | 44.01 |
| BLOOM-560M | 350B | 559M | 27.32 | 24.40 | 46.44 | 44.46 | 36.54 | 28.80 | 62.57 | 53.20 | 42.29 |
| Pythia-410M | 300B | 405M | 29.10 | 24.15 | 51.39 | 59.20 | 40.20 | 29.40 | 66.70 | 53.83 | 46.41 |
| MobileLLM-350M | 1T | 345M | 30.21 | 27.39 | 56.40 | 61.96 | 49.51 | 31.00 | 68.88 | 57.14 | 50.33 |
| SmolLM-360M | 600B | 362M | 33.89 | 36.26 | 70.16 | 55.23 | 53.51 | 37.60 | 71.38 | 57.22 | 54.48 |
| Qwen2-0.5B | 15T | 494M | 31.85 | 28.50 | 55.05 | 61.25 | 49.16 | 32.80 | 69.75 | 57.22 | 50.53 |
| Qwen2.5-0.5B | 17T | 494M | 33.37 | 32.17 | 64.44 | 61.99 | 52.09 | 35.20 | 70.29 | 56.20 | 53.20 |
| EfficientLLM-A | 50B | 457M | 33.09 | 35.92 | 70.50 | 59.85 | 53.16 | 35.00 | 72.69 | 56.27 | 54.77 |
| EfficientLLM-A | 500B | 457M | 34.54 | 38.40 | 72.10 | 62.42 | 56.84 | 40.40 | 73.83 | 57.46 | 57.35 |
| OPT-1.3B | 180B | 1.3B | 29.57 | 30.03 | 57.49 | 56.54 | 53.66 | 32.80 | 72.31 | 59.04 | 51.70 |
| GPT-neo-1.3B | 380B | 1.3B | 30.00 | 25.94 | 56.31 | 61.90 | 48.99 | 33.40 | 71.00 | 54.62 | 50.31 |
| BLOOM-1.1B | 350B | 1.1B | 29.16 | 25.77 | 51.73 | 59.51 | 43.11 | 29.60 | 67.30 | 54.62 | 47.38 |
| Pythia-1B | 300B | 1.0B | 30.14 | 26.96 | 56.86 | 60.04 | 47.15 | 31.20 | 70.29 | 52.88 | 49.34 |
| TinyLlama-1.1B | 3T | 1.1B | 32.30 | 30.29 | 60.40 | 56.85 | 59.13 | 35.80 | 73.07 | 59.04 | 53.51 |
| ShearedLlama-1.3B | 50B | 1.3B | 31.51 | 29.44 | 61.07 | 61.83 | 59.33 | 34.40 | 73.94 | 58.01 | 54.00 |
| OLMo-1B | 2T | 1.2B | 32.03 | 30.72 | 63.55 | 61.38 | 62.86 | 36.40 | 75.35 | 59.35 | 55.66 |
| Llama3.2-1B | – | 1.2B | 36.31 | 31.48 | 65.28 | 63.88 | 63.69 | 37.40 | 74.59 | 60.54 | 56.69 |
| EfficientLLM-A | 50B | 1.1B | 36.71 | 40.36 | 73.61 | 62.39 | 60.24 | 40.20 | 75.19 | 61.25 | 59.03 |
| EfficientLLM-A | 320B | 1.1B | 37.71 | 42.24 | 73.48 | 67.09 | 64.09 | 41.80 | 75.41 | 61.17 | 60.75 |

Figure 4: Win rate of EfficientLLM in the instruction tuning task.

and diagonal Hessian-based metrics into Unified Pruning. We further extend this by adding second-order weight updates (Section 3.3) to support OBC-based methods. Reaching the pruning target takes about 4,000 steps with a batch size of 1M tokens. As shown in Table 2, LLM-Pruner, OBC, and diagonal Hessian metrics improve accuracy by 12.72%, 14.39%, and 12.25%, respectively. According to Appendix B.5, EfficientLLM-B performs better in small scale pretraining, and similar in large scale. For generality, we apply the LLM-pruner metric in main results.

**Ablation Studies.** To evaluate each part of the unified pruning, we decouple into 3 basic designs to improve edge language model pretraining: the auto-designed architecture, pruning, and scalable pruning stage. As shown in Fig. 3, we scale up unified pruning according to Appendix B.3. As shown in Table 3, the pruned architecture, unified pruning, and scaling up pruning with 78B tokens continuously improve 5.82% accuracy. We further scale up finetuning (or continued pretraining) to 50B tokens, as in ShearedLlama, and achieve 7.53% accuracy overall.

**Architecture Robustness.** In Appendix A.1, architectures are stably optimized according to dynamic saliency. In Appendix A.2, we evaluate different pruning trajectories to reach the searched architecture. Once we find the optimal architecture, a different trajectory can achieve the same or better results, which we refer to the generalized Lottery Ticket Hypothesis (Frankle & Carbin, 2018).

## 4.2 MAIN RESULTS

**Edge Language Modeling.** For fair comparison, we collect main streams of edge language models in $100M \sim 1B$ sizes, evaluate in the same conditions (Appendix B.2), and make a benchmark in Table 4. 1) Early edge models including OPT (Zhang et al., 2023b), GPT-neo (Black et al., 2022), Pythia (Biderman et al., 2023), and BLOOM (Le Scao et al., 2023) are direct pretrained in limited tokens and sub-optimal architectures, which largely hinder the performance. By leveraging unified pruning, EfficientLLM achieves both architecture and data efficiency. For instance, EfficientLLM-134M exceeds Pythia-410M by 4.13% average accuracy; EfficientLLM-1.1B with 50B tokens exceeds OLMo-1B, TinyLlama, Llama3.2-1B in accuracy. 2) Compared with the SoTA edge model MobileLLM (Liu et al., 2024b), EfficientLLM-134M exceeds MobileLLM-125M by 5.52% with the large scale model compression. 3) Recent SoTA industrial models scaling up pretraining tokens

Table 5: Comparisons of LLM pruning in Llama-7B. We scale up pruning-aware pretraining to 5B tokens for EfficientLLM. #Tuning donates whether to finetune after pruning. Most works report finetuned results.

| #Ratio | Model | #Tuning | ARC-c | ARC-e | BoolQ | HellaSwag | OBQA | PIQA | WinoGrande | Avg. |
|--------|-------|---------|-------|-------|-------|-----------|------|------|------------|------|
| 50% | MaP | ✓ | 30.63 | 49.32 | 39.69 | 42.49 | 31.40 | 66.81 | 50.67 | 44.43 |
| | MvP | ✓ | 26.79 | 44.07 | 59.94 | 40.98 | 31.80 | 63.06 | 55.64 | 46.04 |
| | WANDA | ✓ | 34.20 | 42.68 | 50.90 | 38.12 | 38.78 | 57.38 | 55.98 | 45.43 |
| | LLM-Pruner | ✓ | 28.24 | 46.46 | 61.47 | 47.56 | 35.20 | 68.82 | 55.09 | 48.98 |
| | LoRAPrune | ✓ | 31.62 | 45.13 | 61.88 | 47.86 | 34.98 | 71.53 | 55.01 | 49.72 |
| | LoRAShear | ✓ | 32.26 | 47.68 | 62.12 | 48.01 | 34.61 | 71.80 | 56.29 | 50.40 |
| | Compresso | ✓ | 27.82 | 48.82 | 60.09 | 39.31 | 33.40 | 66.70 | 51.93 | 46.87 |
| | NutePrune | ✗ | 31.74 | 46.59 | 62.20 | 53.87 | 35.80 | 69.91 | 57.77 | 51.13 |
| | NutePrune | ✓ | 32.17 | 51.68 | 62.26 | 55.88 | 34.40 | 71.00 | 57.54 | 52.13 |
| | EfficientLLM-A | ✗ | 30.80 | 52.15 | 62.29 | 54.70 | 35.20 | 56.75 | 51.89 |
| | EfficientLLM-A | ✓ | **34.04** | **64.81** | **64.83** | **60.12** | **34.60** | **73.88** | **61.48** | **56.25** |
| 70% | LLM-Pruner | ✓ | 24.83 | 39.56 | 47.28 | 31.66 | 28.80 | 60.83 | 50.75 | 40.53 |
| | NutePrune | ✓ | 26.19 | 42.17 | 62.08 | 39.43 | 30.20 | 62.30 | 51.46 | 44.83 |
| | EfficientLLM-A | ✗ | 27.73 | 54.50 | 47.89 | 47.77 | 31.00 | 68.17 | 55.17 | 47.46 |
| | EfficientLLM-A | ✓ | **29.95** | **58.59** | **58.13** | **52.02** | **34.60** | **70.08** | **55.96** | **51.33** |

like Qwen (Yang et al., 2024a;b) and Llama3.2-1B, EfficientLLM-457M and 1.1B outperforms Qwen2.5-0.5B by 4.15% and Llama3.2-1B by 4.06% respectively with limited pretraining data. As shown in Appendix C.1, EfficientLLM-457M achieves higher accuracy while requiring **62×** and **16×** fewer GPU hours than Qwen2.5-0.5B when using 50B and 500B continued pretraining tokens.

**Instruction Tuning.** We finetune EfficientLLM-1.1B and other top-quality open-source base models includes OLMo-1B, ShearedLlama-1.3B, TinyLlama-1.1B and Llama3.2-1B in the same condition. We finetune on the Alpaca dataset (Taori et al., 2023) with 52K instructions for 3 epochs. As shown in Fig. 4, EfficientLLM-1.1B significantly outperforms SoTA baselines, indicating the generalization ability in the supervised finetuning (SFT). More case studies are shown in Appendix D.

**Inference Speed & Quantization.** Most edge devices are non-GPU environments. We deploy edge models using 1, 2, 4, and 8 Intel Xeon @ 2.90GHz CPUs respectively. As shown in Appendix C.2, when using 2 CPUs, EfficientLLM-457M speeds up ×**8.7** and ×**3.2** than MobileLLM-350M and Qwen2.5-0.5B respectively; EfficientLLM-1B speeds up ×**7.3** and ×**1.2** than OLMo-1B and Llama3.2-1B. In Appendix C.3, we further quantize EfficientLLM with 8 bit weights and 8 bit activations (8W8A) using the general OmniQuant (Shao et al., 2023). After 8W8A quantization, EfficientLLM-457M even improves 0.09% and EfficientLLM-1B only drops 0.27% average accuracy.

### 4.3 COMPARISONS WITH LLM PRUNING

We mainly focus on large pruning ratio because it is more practical to achieve highly efficiency based on heavy source LLMs. In Table 5, we scale up pruning-aware pretraining to only 5B tokens. We report both results with or without finetuning after pruning. Because previous works finetune in different settings, we finetune additional 1B tokens if with it. Notice that, even without finetuning, EfficientLLM exceeds all the according baselines. It is shown that existing LLM pruning is impractical in large pruning ratio. By simply scaling up LLM-Pruner metric in pruning-aware pretraining, EfficientLLM-A significantly exceeds SoTA NutePrune 6.5% in 70% ratio without bells and whistles, while NutePrune integrates distillation and additional learnable masks. In 50% ratio, EfficientLLM exceeds LoRAPrune by 2.18% and 6.54% when with and without tuning. Experiments reveal that only scaling up the pruning stage to 5B tokens can achieve much higher performance than previous results, highlighting the importance of scalable pruning methods.

### 5 CONCLUSION

This work primarily advances the edge language model pretraining to exceed the traditional LLM scaling law. Distinguished from almost LLM compression in post-training, this work scales up existing pruning metric in the pretraining stage, promoting LLM compression to the era of pretraining. Technically, minimal parameter groups are defined and optimized by saliency to address scalable target-agnostic pruning. The results reveal that even if vanilla LLM-Pruner can surpass SoTA pruning methods by scaling up and outperform direct pretraining edge models.

**Limitations.** Future work will explore the reasoning and long-context capabilities of edge language models. Since reasoning is more related to CoT data and base models, we exclude it in this work.

ETHICS STATEMENT

All authors have read and agree to adhere to the ICLR Code of Ethics. This paper presents work whose goal is to advance the field of Machine Learning. At present, we do not identify any specific ethical concerns that require special attention beyond standard considerations of fairness, privacy, security, and research integrity.

REPRODUCIBILITY STATEMENT

We release the code in supplemental materials. All of the training data and evaluation methods are publicly available with clear source.

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

# APPENDIX

## A  AUTO-DESIGNED ARCHITECTURES

### A.1  VISUALIZATION

As shown in Fig.5, we visualize the pruning-aware pretraining. We prune SmolLM-1.7B to EfficientLLM-A-457M. In Fig.5 (right), the self-attention parameter groups and FFN parameter groups are iteratively pruned in the initial stage. After 44.49B-token pretraining, the transformer stem parameter groups start to be pruned. This indicates that for the typical human-designed transformer shape, there are more redundant parameters in the attention head and the intermediate size of FFN compared with the transformer stem.

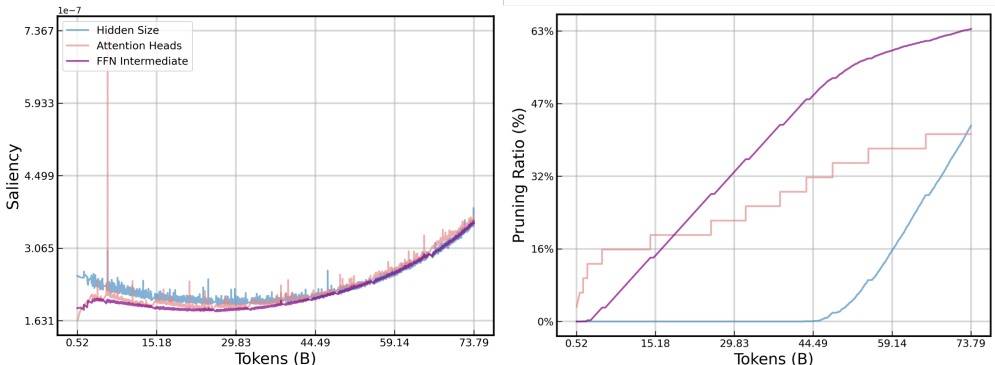

Figure 5: Visualization of pruning-aware pretraining. We plot the saliency of the three pruning types and their pruning ratio in training.

### A.2  ARCHITECTURE ROBUSTNESS

We conduct further experiments to demonstrate how different pruning paths can lead to stable (and generally better) results. We also firstly explore the connection between the architecture robustness and the generalized Lottery Ticket Hypothesis (Frankle & Carbin, 2018), where not only parameter initialization but also architecture initialization satisfies the lottery ticket condition.

We began with 2000 steps of Unified Pruning to obtain an automatically designed model architecture, denoted as A*. We then mark A* as the target and restarted training from scratch, enforcing different pruning paths.

Specifically, we randomly sampled three pruning paths:

i) Prune in the order: Stem → Attention → FFN

ii) Prune in the order: Attention → Stem → FFN

iii) Prune in the order: FFN → Stem → Attention

Table 6: Performance of different pruning trajectories (7 zero-shot average).

| Pruning Traj. | stem_attn_ffn | attn_stem_ffn | ffn_stem_attn | Unified Pruning |
|---|---|---|---|---|
| Avg. (7 zero-shot) | 44.52% | 44.11% | 44.15% | 43.92% |

The experimental results show that, given a known target structure A*, different pruning paths consistently lead to similar or even better performance.

In the classic Lottery Ticket Hypothesis (Frankle & Carbin, 2018), the initialization of model parameters determines whether they will successfully train—winning tickets remain effective regardless of the training process. Similarly, in our experiments, we find that once the optimal target architecture

has been identified, variations in training dynamics (i.e., pruning paths) do not significantly affect the final accuracy, even without fixing specific parameters—only the architecture.

However, this does not imply that scaling up the pruning process is unimportant. On the contrary, it plays a key role in discovering a more accurate target architecture.

## A.3 ARCHITECTURE COMPARISONS

Table 7: Architecture comparisons between EfficientLLM and human-designd models.

| Model | Hidden Size | FFN Intermediate | Attention Heads | Head Dim | Layer |
|---|---|---|---|---|---|
| MobileLLM-125M | 576 | 1536 | 9 | 64 | 30 |
| EfficientLLM-A-134M | 757 | 966 | 5 | 64 | 32 |
| MobileLLM-350M | 960 | 2560 | 15 | 64 | 32 |
| Qwen2/2.5 | 896 | 4864 | 14 | 64 | 24 |
| EfficientLLM-A-457M | 1195 | 3006 | 19 | 64 | 24 |
| MobileLLM-1B | 1280 | 3584 | 20 | 64 | 54 |
| ShearedLlama-1.3B | 2048 | 5504 | 16 | 128 | 24 |
| OLMo-1B | 2048 | 8192 | 16 | 128 | 16 |
| Llama3.2-1B | 2048 | 8192 | 32 | 64 | 16 |
| EfficientLLM-A-1.1B | 2048 | 4870 | 24 | 64 | 24 |

Table 8: Architectures in different pruning metrics to scale up by pruning-aware pretraining. We compare the approximate 460M model size. "x1" indicates that the number of gradient descent steps and pruning steps in each iteration are 1:1.

| Model | Hidden Size | FFN Intermediate | Attention Heads | Head Dim | Layer |
|---|---|---|---|---|---|
| LLM-Pruner x1 (Ma et al., 2023) | 1169 | 3082 | 19 | 64 | 24 |
| OBC x1 (Frantar & Alistarh, 2022) | 1131 | 3258 | 19 | 64 | 24 |
| Diag-Hess x1 (LeCun et al., 1989) | 1963 | 1542 | 12 | 64 | 24 |

As shown in Table 7, we compare the auto-designed architectures by saliency via pruning and the best practices of human design, including MobileLLM and Qwen2/2.5-0.5B, OLMo-1B, ShearedLlama-1.3B. In EfficientLLM, the pruning ratio of hidden-size is smaller than attention heads and FFN intermediate channels driven by saliency.

As shown in Table 8, we compare the influence of different pruning metrics, including the classic LLM-Pruner (Ma et al., 2023), OBC (Frantar & Alistarh, 2022), and Diag-Hess (LeCun et al., 1989). The Diag-Hess only uses the second-order term in Eq.9, which applies the diagonal of the Hessian matrix for approximate calculation.

## A.4 CLUSTER ATTENTION

Pruning-aware pretraining could structurally prune the Group Query Attention (GQA) (Ainslie et al., 2023), which is usually applied for KV cache compression in LLMs. When the source model applies GQA, there are different cases in pruning:

- in all of the following cases, the query attention heads is the same in each layer, and the same as the self-attention operation. The difference is how to share key and value for queries.

- As shown in Fig.8, if all queries corresponding to a key and value are pruned, then the key and value are also pruned.

- If a part of the query corresponding to a key and value is pruned, then the key and value are retained. This eventually forms cluster attention.

We plot an example of EfficientLLM-A-134M in Fig.8. And the source models of EfficientLLM-457M and EfficientLLM-1.1B do not apply GQA.

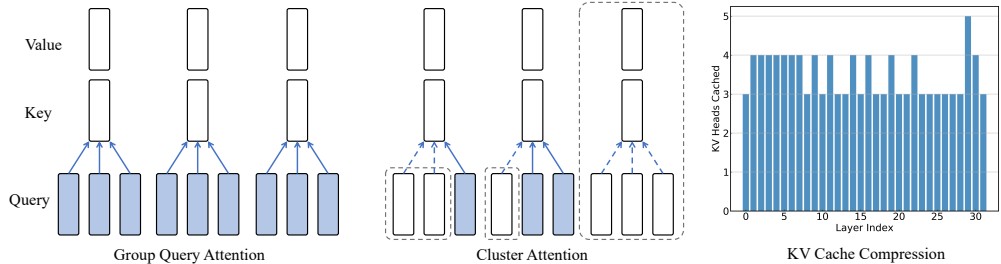

Figure 6: Group Query Attention (GQA) pruning. In the case of GQA, cluster attention can be obtained through pruning. After pruning, the number of query heads is the same in each layer, and the cluster attention compresses the KV Cache.

## B  TRAINING AND EVALUATION DETAILS

### B.1  TRAINING

Our training code and models will be fully open-sourced on GitHub and Huggingface. Detailed hyperparameters are shown in Table 9. Note that iterations determine the number of tokens in pruning-aware pretraining to achieve the target model size, which is not directly defined. It can be adjusted through batch size and the pruning frequency in each iteration.

Table 9: Hyper-parameters in pruning-aware pretraining and continued pretraining stages.

| Model | #Tokens | Learning Rate | WarmUp Steps | Batchsize | Text Length | #GPU |
|---|---|---|---|---|---|---|
| Pruning-134M | 50.3B | $2 \times 10^{-3}$ | 500 | 2 M | 2048 | 32 |
| Continued Pretrain-134M | 500B | $2 \times 10^{-3}$ | 10000 | 1 M | 2048 | 32 |
| Pruning-457M | 72.1B | $5 \times 10^{-4}$ | 500 | 1 M | 2048 | 32 |
| Continued Pretrain-457M | 50B/500B | $2 \times 10^{-3}$ | 10000 | 1 M | 2048 | 40 |
| Pruning-1.1B | 36.7 | $5 \times 10^{-4}$ | 500 | 1 M | 2048 | 32 |
| Continued Pretrain-1.1B | 50B/500B | $5 \times 10^{-4}$ | 10000 | 1 M | 2048 | 64 |

### B.2  EVALUATION

- **MMLU**: According to Datacomp-lm (Li et al.) (Appendix G of Datacomp-lm) and SmolLM (Allal et al., 2024), taking into account the log probabilities of complete answer sequences in MMLU is more related to weaker model performance, such as edge language models. Following SmolLM (Allal et al., 2024), we apply the Lighteval-v0.7.0 (Fourrier et al., 2023) to evaluate MMLU zero-shot performance.

- **Common Sense Reasoning**: Follow most of recent works (Xia et al., 2023; Ma et al., 2023; Li et al., 2024), we apply the widely used lm-evaluation-harness package (Gao et al., 2024) to evaluate zero-shot common sense reasoning tasks. To avoid different results introduced by different versions. We evaluate all the benchmarks with the 0.4.3 version. However, some previous works evaluate in older version 0.3.0, and we evaluate with the same version in Table 5, 50% pruning ratio. Finally, all the versions are the same.

### B.3  SCALABILITY OF UNIFIED PRUNING-AWARE PRETRAINING

In Fig. 3, we evaluate the scalability of pruning-aware pretraining. According to Eq. 7, we set the ratio of pruning steps to gradient descent steps to 4:1, 2:1, 1:1, and 1:9 in a iteration, respectively. When the target model size is reached, the pruning-aware pretraining requires 2.5B, 4.5B, 8.4B, and 72.1B tokens of pretraining, respectively. Fig. 3 indicates that scaling up pruning-aware pretraining continuously improves pruning performance. By scaling up LLM pruning during pretraining, the upper boundary of LLM compression can be extended.

## B.4 COMPARISON WITH THE SOURCE MODEL

Table 10: Comparison between SmolLM-1.7B and EfficientLLM-A-1.1B.

| Model | #Params | ARC-c | ARC-e | BoolQ | HellaSwag | OBQA | PIQA | WinoGrande | Avg. |
|---|---|---|---|---|---|---|---|---|---|
| SmolLM-1.7B | 1.7B | **46.16** | **76.60** | 65.99 | **65.74** | **42.00** | **75.95** | 60.14 | **61.80** |
| EfficientLLM-A-1.1B | 1.1B | 42.24 | 73.48 | **67.09** | 64.09 | 41.80 | 75.41 | **61.17** | 60.75 |

## B.5 COMPARISONS BETWEEN EFFICIENTLLM-A AND B

As shown in Table 11, EfficientLLM* indicates pruning Llama2-7B to 1.3B with RedPajama dataset; EfficientLLM indicates pruning SmolLM-1.7B to 457M in main results. EfficientLLM-B exceeds EfficientLLM-A in small scale pretraining, while their results become similar in large scale pretraining. We finally choose EfficientLLM-A to scale up.

Table 11: Comparisons between EfficientLLM-A and B for the second-order weight updating.

| Model | Tokens | ARC-C | ARC-E | BoolQ | HS | OBQA | PIQA | WG | Avg. |
|---|---|---|---|---|---|---|---|---|---|
| EfficientLLM*-A | 6B | 27.30 | 56.44 | 57.58 | 50.03 | 31.00 | 69.10 | 54.70 | 49.45 |
| EfficientLLM*-B | 6B | 28.58 | 56.90 | 62.42 | 49.81 | 32.20 | 68.93 | 55.49 | 50.62 |
| EfficientLLM-A | 572B | 38.40 | 72.10 | 62.42 | 56.84 | 40.40 | 73.83 | 57.46 | 57.35 |
| EfficientLLM-B | 572B | 39.59 | 71.68 | 62.39 | 57.21 | 39.60 | 73.50 | 57.70 | 57.38 |

## C TRAINING AND INFERENCE EFFICIENCY

### C.1 TRAINING EFFICIENCY

As shown in Table 12, we evaluate pretraining speed under the same environment, as shown in the table. With pruning-aware pretraining, training EfficientLLM-457M requires x16 to x62 times fewer GPU hours compared to Qwen2.5-0.5B, while achieving higher accuracy.

Table 12: Comparison of pre-training and pruning costs in GPU hours. PT indicates pretraining.

| Model | PT Tokens | GPU Hours | Pruning | GPU Hours | Continued PT | GPU Hours | Total Hours | Acc. (%) |
|---|---|---|---|---|---|---|---|---|
| Qwen2.5-0.5B | 17T | 199467 | – | – | – | – | 199467 | 53.20 |
| EfficientLLM-457M | – | – | 72.1B | 2166 | 50B | 1018 | 3184 | 54.77 |
| EfficientLLM-457M | – | – | 72.1B | 2166 | 500B | 10178 | 12344 | 57.35 |

### C.2 INFERENCE EFFICIENCY

As shown in Table 13, we deploy EfficientLLM on non-GPU devices. We deploy on 1, 2, 4, and 8 Intel Xeon @ 2.90GHz CPUs respectively. Compared with SoTA edge models, EfficientLLM achieves both higher inference speed (ms/token) and average zero-shot accuracy.

Table 13: Inference latency (in milliseconds) and accuracy on different models.

| Model (ms) | 1 CPU | 2 CPUs | 4 CPUs | 8 CPUs | Acc. (%) |
|---|---|---|---|---|---|
| MobileLLM-350M | 132.37 | 81.26 | 53.99 | 41.95 | 51.30 |
| Qwen2.5-0.5B | 54.51 | 30.08 | 19.87 | 13.54 | 53.20 |
| EfficientLLM-457M | 15.93 | 9.38 | 5.67 | 4.07 | 57.35 |
| OLMo-1B | 352.92 | 186.39 | 101.81 | 65.12 | 55.66 |
| Llama3.2-1B | 51.45 | 30.17 | 16.59 | 10.33 | 56.69 |
| EfficientLLM-1B | 49.77 | 25.52 | 16.00 | 10.32 | 60.75 |

## C.3 QUANTIZATION

We applied OmniQuant to perform 8-bit weight and activation quantization (8W8A) on EfficientLLM. The experimental results show that 8-bit quantization only causes minor disturbances to the results, further proving the effectiveness of the model acceleration.

Table 14: Performance comparison of models with different bit-widths.

| Model | #Bits | ARC-C | ARC-E | BoolQ | HS | OBQA | PIQA | WG | Avg. |
|---|---|---|---|---|---|---|---|---|---|
| Qwen2.5-0.5B | bf16 | 32.17 | 64.44 | 61.99 | 52.09 | 35.20 | 70.29 | 56.20 | 53.20 |
| EfficientLLM-A-457M | bf16 | **38.40** | 72.10 | 62.42 | 56.84 | **40.40** | **73.83** | 57.46 | 57.35 |
| EfficientLLM-A-457M-Quant | 8W8A | 38.23 | **72.14** | **63.18** | **56.90** | 40.00 | 73.67 | **57.93** | **57.44** |
| Llama3.2-1B | bf16 | 31.48 | 65.28 | 63.88 | 63.69 | 37.40 | 74.59 | 60.54 | 56.69 |
| EfficientLLM-A-1B | bf16 | **42.24** | **73.48** | **67.09** | **64.09** | 41.80 | **75.41** | **61.17** | **60.75** |
| EfficientLLM-A-1B-Quant | 8W8A | 41.64 | 73.32 | 66.51 | 63.98 | 42.40 | 75.24 | 60.30 | 60.48 |

# D CASE STUDY IN ALPACA FINETUNING

In the instruction finetuning task, we finetune EfficientLLM-A-1.1B, OLMo-1B, ShearedLlama-1.3B, and Llama3.2-1B in the Alpaca dataset. And then, we evaluate using the Alpaca-Eval. We sample some instructions in Alpaca-Eval and make comparisons as follows:

---

**Input:**

I want to eventually become a plant/manufacturing/production manager, and eventually a Chief Operating Officer. What sort of experience and post-graduate education path would you recommend? Please provide non-obvious and atypical career paths leading to such positions, on top of the classic ones.

---

**Output** (EfficientLLM-A-1.1B):

To become a plant/manufacturing/production manager and eventually a Chief Operating Officer, you will need to have a combination of experience and post-graduate education.

For experience, you can start by working in a production or manufacturing role. This will give you the opportunity to gain hands-on experience in the industry and learn about the day-to-day operations of a production or manufacturing facility. You can also gain experience by working in a related field, such as engineering, logistics, or supply chain management.

For post-graduate education, you can pursue a degree in a related field, such as industrial engineering, operations management, or supply chain management. You can also pursue a master's degree in a related field, such as operations management or supply chain management. Additionally, you can pursue professional certifications, such as the Certified Production and Inventory Management Professional (CPIMP) or the Certified Supply Chain Professional (CSCP).

In addition to experience and post-graduate education, you will also need to have excellent communication, organizational, and problem-solving skills. You will also need to be able to lead and motivate a team, as well as be able to think strategically and make decisions quickly.

Overall, to become a plant/manufacturing/production manager and eventually a Chief Operating Officer, you will need to have a combination of experience and post-graduate education, as well as excellent communication, organizational, and problem-solving skills. Additionally, you will need to be able to lead and motivate a team, as well as be able to think strategically and make decisions quickly.

---

**Output** (OLMo-1B):
To become a plant/manufacturing/production manager, you will need at least a Bachelor's degree in a relevant field, such as industrial engineering, manufacturing engineering, or production management. To become a Chief Operating Officer, you will need at least a Master's degree in a relevant field, such as business administration, management, or marketing. To become a plant/manufacturing/production manager, you will need at least a PhD in a relevant field, such as industrial engineering, manufacturing engineering, or production management. Additionally, to become a plant/manufacturing/production manager, you may also want to obtain a certification such as the Certified Production Manager (CPM) or Certified Quality Manager (CQM) to demonstrate your expertise in the field.

**Output** (ShearedLlama-1.3B):
To become a plant/manufacturing/production manager, you will need to have a bachelor's degree in a relevant field such as engineering, business, or manufacturing. After that, you can pursue a master's degree in a related field such as operations management, manufacturing engineering, or production and operations management. Additionally, you may want to pursue post-graduate education in areas such as project management, lean manufacturing, or supply chain management.
For a Chief Operating Officer position, you will need to have a bachelor's degree in a relevant field such as engineering, business, or manufacturing. After that, you can pursue a master's degree in a related field such as operations management, manufacturing engineering, or production and operations management. Additionally, you may want to pursue post-graduate education in areas such as project management, lean manufacturing, or supply chain management.
Overall, to become a plant/manufacturing/production manager or Chief Operating Officer, you will need to have a bachelor's degree in a relevant field such as engineering, business, or manufacturing, and then pursue a master's degree in a related field such as operations management, manufacturing engineering, or production and operations management. Additionally, you may want to pursue post-graduate education in areas such as project management, lean manufacturing, or supply chain management.

**Output** (Llama3.2-1B):
To become a plant/manufacturing/production manager and eventually a Chief Operating Officer, it is recommended to have a degree in engineering, business, or a related field, as well as relevant post-graduate education and experience. Examples of post-graduate education include an MBA, a Master's degree in engineering, or even a PhD in a related field. Relevant experience could include working in a manufacturing or production role, managing a team of engineers, or leading a plant or production department.

**Input:**
Why so many people like film photography when digital camera and mobile phone are more convenient and generate higher quality of photos?

**Output** (EfficientLLM-A-1.1B):
Many people still prefer film photography because of its unique aesthetic qualities. Film photography allows for more creative control, as the photographer can adjust the ISO, shutter speed, and aperture to achieve the desired effect. Additionally, film photography produces a unique look and feel that cannot be replicated with digital cameras and mobile phones. Film photography also allows for more experimentation, as the photographer can adjust the ISO, shutter speed, and aperture to achieve the desired effect. Finally, film photography can be a more personal and intimate experience, as it allows the photographer to capture moments in a way that digital cameras and mobile phones cannot.

**Output** (OLMo-1B):
Film photography has many advantages compared to digital camera and mobile phone, such as higher quality of photos, more creative control, and the ability to capture moments that would otherwise go unnoticed. Additionally, film photography requires more time and patience to develop the film, which adds an element of adventure and creativity to the process. Finally, many people find the physical process of developing film to be aesthetically pleasing, while digital camera and mobile phone processing can often produce photos with a less-than-desirable outcome.

**Output** (ShearedLlama-1.3B):
People like film photography because it allows for more creative control. With film photography, the photographer has the ability to adjust the exposure, focus, and other settings to create the desired effect. Digital cameras and mobile phones do not offer this level of creative control, making it difficult to achieve the desired effect. Additionally, film photography requires more thought and preparation than digital camera and mobile phone photography, making it more enjoyable and rewarding.

**Output** (Llama3.2-1B):
People like film photography because it produces higher quality of photos than digital cameras and mobile phones. Film photography is more expensive than digital cameras and mobile phones, but the quality of photos produced by film is much better than those produced by digital cameras and mobile phones. Additionally, film photography requires more skill and patience to take good photos, which many people enjoy.

