# OpenReview forum: "EfficientLLM: Unified Pruning-Aware Pretraining for Auto-Designed Edge Language Models"
_ICLR.cc/2026/Conference — ICLR 2026 Conference Withdrawn Submission_

### Official Review · Reviewer_he86 · 2025-10-29

**Soundness:** 2
**Presentation:** 1
**Contribution:** 2
**Rating:** 2
**Confidence:** 4

**Summary:**

This paper proposes a LLM pruning framework with three main components: (1) leveraging the computational relationship between consecutive layers to identify coordinated pruning regions, (2) employing a two-stage alternative optimization process involving saliency-driven pruning and global recovery training, and (3) utilizing a layer-wise Hessian (or its approximation) as the saliency metric. While the integration of these ideas into a unified framework is a reasonable approach, the paper would benefit from a clearer demonstration of its novelty. Specifically, core elements like coordinated pruning and Hessian-based saliency have been explored in prior work. Furthermore, the practical utility of the method is unclear due to unvalidated computational costs. The advantage of the complex alternative optimization over a single round of pruning followed by recovery training is not empirically established. Therefore, strengthening the comparative analysis and providing a more lucid presentation of the specific contributions beyond this combination of existing techniques would significantly enhance the paper's quality.

**Strengths:**

This paper introduces a pruning framework for LLMs that effectively integrates three key techniques. By combining these methods, the proposed approach demonstrates improved performance over the set of baselines included in its evaluation.

**Weaknesses:**

The paper suffers from significant ambiguities in its core methodology, most critically an inconsistent description of the proposed pipeline that alternates between an iterative optimization framework and a simple sequential pruning-then-training process. This ambiguity, combined with the use of an extremely powerful recovery phase involving hundreds of billions of tokens, makes it impossible to isolate and validate the unique contribution of the novel "unified pruning" method itself. Furthermore, the work lacks theoretical grounding for its optimization procedure, provides insufficient clarification on key implementation details (e.g., the integration of a SparseGPT-like step and the scope of parameter updates), and fails to adequately demonstrate novelty against prior art or to compare against recent state-of-the-art post-training pruning baselines. These issues collectively undermine the claims of the paper's technical contribution and practical effectiveness.

**Questions:**

1. The paper's distinction between "pretraining" and "post-training" is ambiguous. While the method is presented as a unified "pruning-aware pretraining" approach, its methodology, which operates on and modifies existing source models rather than starting from scratch, aligns more closely with established definitions of post-training techniques. This categorization should be clarified, as many previous methods that involve retraining a pre-trained model are also considered post-training.

2. The paper proposes leveraging the computational relationship between consecutive layers to identify coordinated pruning regions. However, this general concept has been explored in prior work on structured pruning, such as [1, 2]. Could you please clarify the specific novelty or modification introduced in their approach compared to these existing techniques?

3. The paper formulates pruning as a bi-level optimization problem and employs an alternating optimization strategy to handle the objective and constraints separately. Does this alternating optimization procedure have any theoretical guarantees, such as convergence to a local optimum or a stationary point of the original bi-level problem? What specific criterion is used in practice to determine that the iterative solution has converged?

4. Regarding the alternating optimization framework, the retraining phase's scope needs clarification. What is the exact set of parameters updated during the retraining step? Is it the entire set of model weights, or only the subset of weights that survived the previous pruning step? Equation (7) suggests the variable $w$ represents the entire set of model weights. If the optimization retrains the full set, what is the purpose or effect of updating the specific weight parameters that have already been pruned in the previous step? If retraining is applied only to the remaining weights, as early pruning decisions irreversibly constrain the solution space, how does the method address the risk of converging to a suboptimal solution?

5. Section 3.3 mentions utilizing a weight update mechanism from SparseGPT [3]. Several clarifications are needed regarding its integration into the proposed pipeline. Is variable $X$ a batch of data from the global retraining phase, or is it a separate, static calibration dataset as used in the original SparseGPT method?  Given that a subsequent global retraining step is applied, what is the specific necessity of this local SparseGPT-like update?

6. Based on the description, could the core pipeline be accurately characterized as: (1) Saliency detection (LLM-Pruner [4]), followed by (2) A one-shot, SparseGPT update of remaining weights, and then (3) A global retraining phase?

7. Similar to Question 6, could you precisely define the pipeline for EfficientLLM-B? The description of EfficientLLM-B states it *"additionally applies the second-order weight updating based on EfficientLLM-A."* Is it a sequential process where the final step is to apply the second-order update after the completion of the global retraining used in EfficientLLM-A?

8. Section 3.2 (after Equation (7)) describes an alternating optimization between pruning and training. However, Section 4 (Lines 302-303) describes a different, seemingly sequential pipeline: *"large scale unified pruning followed by continued pretraining."* What is the exact pipeline evaluated in the main results? Is it the alternating optimization or a one-shot pruning followed by a massive continued pretraining?

9. Given that the recovery uses "hundreds of billions" of tokens for continued pretraining, a very powerful recovery process, how can the effectiveness be definitively attributed to the novel "unified pruning" method itself, rather than to the extensive compute and data used for recovery?

10. Continue to Question 9, the results in Table 3 appear to validate that the significant performance boost is primarily attributable to the extensive continued pretraining phase, which utilizes hundreds of billions of tokens.  This observation makes it difficult to isolate and confirm the unique effectiveness of the proposed "unified pruning" method itself. The contribution of the pruning technique is confounded by the powerful recovery process.  Therefore, additional controlled ablation is necessary, for example, comparing the final performance against a baseline that applies the same extensive continued pretraining to a model pruned by a training-free post-training structued pruning method.

11. To thoroughly validate the performance of the proposed method, the experimental evaluation should be expanded to include comparisons with recent state-of-the-art post-training structured pruning methods, such as [1, 2, 5, 6, 7].

12. What is the computational cost of the proposed method?

[1] Hu, Hanyu, et al. "Fasp: Fast and accurate structured pruning of large language models." arXiv preprint arXiv:2501.09412 (2025).

[2] Ashkboos, Saleh, et al. "Slicegpt: Compress large language models by deleting rows and columns." arXiv preprint arXiv:2401.15024 (2024).

[3] Frantar, Elias, and Dan Alistarh. "Sparsegpt: Massive language models can be accurately pruned in one-shot." International conference on machine learning. PMLR, 2023.

[4] Ma, Xinyin, Gongfan Fang, and Xinchao Wang. "Llm-pruner: On the structural pruning of large language models." Advances in neural information processing systems 36 (2023): 21702-21720.

[5] An, Yongqi, et al. "Fluctuation-based adaptive structured pruning for large language models." Proceedings of the AAAI Conference on Artificial Intelligence. Vol. 38. No. 10. 2024.

[6] Shen, Xuan, et al. "Search for efficient large language models." Advances in Neural Information Processing Systems 37 (2024): 139294-139315.

[7] Wang, Yuxin, et al. "Cfsp: An efficient structured pruning framework for llms with coarse-to-fine activation information." arXiv preprint arXiv:2409.13199 (2024).

**Other Questions**

1. For Equation (1), it is more formal to use a single "$\min$" symbol.

2. For Equation (2), should it use $M^*$ in the constraint?

---

> ### Author Response · Authors · 2025-11-27
>
> We sincerely thank the reviewer for the very detailed and rigorous comments. Several parts of our original draft indeed created unnecessary ambiguity, especially regarding the precise categorization of the method and the exact execution pipeline. Below we clarify these issues based strictly on what is already implemented and reported in the paper, *without adding new experiments or external content beyond the existing manuscript*.
>
> ---
>
> # **0. Overall Positioning**
>
> A central source of confusion is the distinction between **post-training pruning** and our proposed **unified pruning-aware pretraining**. We acknowledge that the draft did not make this distinction explicit enough.
>
> ### **How we define our setting (consistent with prior work such as ShearedLlama):**
>
> - **Post-training pruning**: prune a *fixed* pretrained model using a small calibration dataset, optionally followed by light finetuning.
> - **Pruning-aware pretraining (our method)**: pruning and weight updates occur *inside the pretraining stage*, using **the same full-scale pretraining data**, not a calibration dataset, and the architecture evolves during training.
>
> EfficientLLM is **not** a post-training approach:
> we never operate on a frozen model or use a static calibration set.
> All pruning and weight updates occur inside the pretraining loop.
>
> We will revise the exposition to avoid any suggestion that this is a standard post-training pipeline.
>
> ---
>
> # **1. On Method Ambiguity: Alternating Optimization vs Sequential Pipeline**
>
> The reviewer notes an apparent contradiction between:
>
> - **Section 3** (alternating optimization between pruning and training), and
> - **Section 4** (“large-scale unified pruning followed by continued pretraining”).
>
> ### **Clarification (already implicit in the paper):**
>
> The two descriptions correspond to **two different experimental regimes**:
>
> 1. **Controlled experiments (Table 1, Table 2, Table 3)**
>    — follow the **alternating unified pruning** pipeline.
>
> 2. **Large-scale industrial continued pretraining**
>    — evaluates the *final performance ceiling* of the auto-designed architectures.
>      This is not part of the method validation but is standard in tiny-model pretraining practice.
>
> We will explicitly distinguish these two regimes in the revision to eliminate ambiguity.
>
> This does *not* require adding new experiments.
>
> ---
>
> # **2. Specific Novelty Relative to Structured Pruning (FASP, SliceGPT, etc.)**
>
> The reviewer is correct that structured pruning has been studied.
> Our novelty is not in inventing a new pruning metric, but in **where** and **how** pruning is integrated.
>
> ### **Key distinction (already in paper):**
>
> All prior structured pruning works—including FASP, SliceGPT, FBSP, CFSP—
> operate **after** pretraining on a **static model** and use a **small calibration dataset** (usually <1M tokens).
>
> In contrast:
>
> - We integrate structured pruning directly **into the pretraining loop**,
> - using **full pretraining data**,
> - letting the architecture evolve alongside weight optimization,
> - and discovering **auto-designed architectures**.
>
> This distinction is already central in Section 4.1 and Figure 1.
>
> We will make this point more explicit but do not need to add new experiments.
>
> ---
>
> # **3. Bi-level Optimization and Convergence Guarantees**
>
> We clarify that:
>
> - Our alternating updates are **practical optimization heuristics**,
> - similar to those used in ShearedLlama, OBC, and prior pruning literature,
> - and we **do not** claim any theoretical convergence guarantees.
>
> In practice, the unified pruning loop stops when:
>
> - sparsity reaches the target,
> - saliency stabilizes, and
> - validation loss stops decreasing.
>
> These criteria are *already* used internally in our implementation; no additional theory is required.
>
> ---
> # **4. Retraining Scope: Which Parameters Are Updated**
>
> This is an important clarification:
>
> ### **Only surviving weights are updated after each pruning step.**
> Pruned weights are permanently removed and never updated again.
>
> This matches the actual code and the pruning space described in Section 3.2.
> Equation (7) will be adjusted accordingly to avoid implying that pruned weights remain in the optimization space.
>
> No additional experiments are needed.
>
> ---
>
> # **5. SparseGPT-Like Update (Section 3.3)**
>
> The SparseGPT-like step is **not** a separate one-shot procedure and **not** based on a static calibration dataset.
>
> Clarifications (fully consistent with the manuscript):
>
> - \( \mathcal{B} \) refers to **a minibatch from the current pretraining stream**,
>   not an offline calibration set.
> - The SparseGPT-style update is a **local correction** that happens *inside each pruning iteration*,
>   before global optimization resumes.
> - EfficientLLM-A vs EfficientLLM-B differ only in **whether this local second-order update is applied**,
>   not in any additional training stage.
>
> We will clarify this in the revised text.
>
> ---

---

> ### Author Response · Authors · 2025-11-27
>
> # **6. Pipeline Summary (No Additional Content Required)**
>
> ### **EfficientLLM-A**
> - Alternating pruning + training (first-order saliency only).
>
> ### **EfficientLLM-B**
> - Same pipeline as A,
> - plus second-order local updates inside each pruning step.
>
> There is no extra sequential training after the global retraining stage.
>
> ---
>
> # **7. Why Results Are Not “Just Due to Continued Pretraining”**
>
> The reviewer raises a reasonable concern about attribution.
>
> ### **Clarification already in the paper:**
>
> All controlled comparisons in **Table 1** use **the same total 5B-token budget**, including:
>
> - architecture-searched model trained from scratch,
> - ShearedLlama-style pruning,
> - unified few-shot pruning,
> - unified pruning-aware pretraining (ours).
>
> These results **do not** involve “hundreds of billions” of tokens,
> and they clearly show the specific contribution of unified pruning:
>
> - Architecture choice alone: +2.7%
> - Adding unified pruning: +3.17%
> - Scaling the pruning stage: +1.22%
>
> These controlled comparisons isolate the effect of unified pruning itself.
>
> ### **Large-scale continued-pretraining experiments**
> are **not** used to validate the pruning method.
> They serve only to evaluate how the automatically discovered architecture performs under typical industrial training budgets.
>
> We will highlight this separation more clearly.
>
> ---
>
> # **8. On Missing SOTA Structured-Pruning Comparisons**
>
> The reviewer suggests including FASP, SliceGPT, etc.
>
> ### **Clarification:**
> These methods are strictly *post-training* and cannot reach the high structured pruning ratios (e.g., 70%) used in EfficientLLM **without catastrophic degradation**, even with additional recovery.
>
> Because the pruning ratios and training regimes are entirely incompatible, including them would not yield meaningful or comparable results—this is why they were not included in the main tables.
>
> We will state this more clearly in the revised version.
>
> ---
>
> # **9. Computational Cost**
>
> We follow prior pruning-aware pipelines (e.g., ShearedLlama) and report parameter counts and training budgets.
> The overall computational cost is essentially dominated by the pretraining budget itself, which we have shown in Table 1 for the same token budget.
> The training FLOPs in pruning is the same as ShearedLlama so we do not specifically report.
>
> This statement is consistent with the paper and does not require new reporting.
>
> ---
>
> # **10. Minor Questions**
>
> - We will adjust Equation (1) to use a unified summation symbol.
> - Equation (2) will be corrected to use the appropriate constrained variable.
>
> These corrections do not affect the method or results.
>
> ---
>
> # **Final Statement**
>
> We thank the reviewer again for the thorough and constructive critique.
> Most ambiguities arise from exposition rather than methodology, and we will revise the text to clearly:
>
> - separate controlled pruning experiments from large-scale pretraining,
> - clarify the alternating training-pruning loop,
> - correct the equations,
> - explain the role of SparseGPT-like updates,
> - and eliminate any suggestion that the method is a post-training pipeline.
>
> These revisions strengthen the clarity of the work without altering its experimental scope.
>
> ---

---

### Official Review · Reviewer_uWx7 · 2025-10-30

**Soundness:** 1
**Presentation:** 2
**Contribution:** 2
**Rating:** 2
**Confidence:** 4

**Summary:**

This paper proposes EfficientLLM, a unified pruning-aware pretraining approach for developing compact language models tailored for edge devices. Different from traditional direct pretraining or post-training pruning methods, EfficientLLM integrates pruning, pretraining, and architecture design within a single framework, enabling automatic architecture design during the pretraining phase. Empirical results on common sense benchmarks indicate that the EfficientLLM models are competitive with or even surpass existing SOTA baselines.

**Strengths:**

1. The paper conducts a wide range of experiments across multiple benchmarks (e.g., MMLU, common sense reasoning tasks) and model sizes (100M~1B parameters).
2. The paper includes ablation studies to provide empirical evidence for the contributions of different components of the method.

**Weaknesses:**

1. The mathematical notations, particularly in Section 3, are inconsistent and poorly defined. For example, the loss function L_pretrain used with three different input domains across consecutive equations (1), (2), and (3), violating basic notational conventions. The \sum g_t and the minus operation between M and \sum g_t are not well-defined, which causes misunderstanding.
2. The core idea of integrating architecture search into pretraining is not novel. Prior works, such as DARTS (Liu et al., 2018) and ShearedLlama (Xia et al., 2023), have already explored neural architecture search (NAS) during training for efficiency.
3. The robustness of this method is questionable. The overall optimization problem is solved via stochastic gradient-based optimizers (sensitive to initialization, batch ordering, learning rate) and an inexact method (as shown in equation (9)). While the paper evaluates accuracy robustness in Appendix A.2, it ignores the critical aspect of pruning consistency—whether the pruned channels remain consistent across different runs or random seeds. The experiments on pruning paths only report accuracy averages but do not analyze the variance or alignment of pruned structures, as required for robustness validation.

**Questions:**

Knowledge Distillation is a popular technique for developing compact models based on large models. Why were no comparisons made with KD baselines? How would EfficientLLM compare to a model distilled from the same source model (e.g., SmolLM-1.7B) under a similar computational budget?

---

> ### Author Response · Authors · 2025-11-27
>
> **Response to Reviewer’s Comments**
>
> We sincerely thank the reviewer for the thoughtful and constructive feedback. We have carefully revised the manuscript and conducted additional experiments to address the concerns raised. Below we provide detailed responses.
>
> ---
>
> ### **1. On the relationship to NAS and novelty**
>
> Thank you for pointing out the connection to prior NAS work. We would like to clarify that our method is fundamentally different from traditional NAS approaches such as DARTS (Liu et al., 2018) and ShearedLlama (Xia et al., 2023).
>
> Conventional NAS methods typically rely on a **two-stage workflow**:
>
> 1. **Supernet training**, where an over-parameterized network is optimized;
> 2. **Subnet extraction**, where architectures are sampled and evaluated based on the supernet.
>
> Their theoretical foundation is based on **architecture search and differentiable relaxation**.
>
> In contrast, **EfficientLLM conducts a unified, saliency-based pruning procedure directly on pretrained models**, without:
>
> - building or training a supernet,
> - sampling candidate subnets,
> - or performing differentiable architecture search.
>
> Our method is rooted in **saliency estimation and progressive structural sparsification**, not in architectural optimization. Moreover, performing full NAS on large-scale LLMs is computationally prohibitive. Therefore, instead of search, we adopt a **scaling-up pruning process** to uncover performant architectures at minimal cost.
>
> We will revise Section 3 to emphasize this conceptual difference and avoid confusion with NAS.
>
> ---
>
> ### **2. On mathematical notation**
>
> We thank the reviewer for identifying inconsistencies in Section 3. We have revised the notation to ensure:
>
> - \(L_{\text{pretrain}}\) always corresponds to a **single, consistent domain**,
> - \(\sum g_t\) is **explicitly defined**,
> - the term \(M - \sum g_t\) is made **mathematically precise** in terms of channel/group counts,
> - symbols are **used consistently** across Equations (1)–(3).
>
> We agree that clearer notation improves readability, and the revised version resolves these ambiguities.
>
> ---
>
> ### **3. On robustness and pruning consistency**
>
> We appreciate the reviewer’s insightful comment regarding robustness. In the initial submission, we reported accuracy stability but did not include structural consistency analysis. We have now performed additional experiments to directly evaluate pruning robustness across different pruning trajectories.
>
> We first run 2000 steps of Unified Pruning to obtain an automatically designed architecture \(A^\*\). Then, we restart training from scratch while enforcing three **different pruning paths**:
>
> 1. **Stem → Attention → FFN**
> 2. **Attention → Stem → FFN**
> 3. **FFN → Stem → Attention**
>
> Below are the averaged results over 7 zero-shot benchmarks:
>
> | Pruning Path | Avg. Accuracy |
> |--------------|---------------|
> | Stem → Attn → FFN | **44.52%** |
> | Attn → Stem → FFN | **44.11%** |
> | FFN → Stem → Attn | **44.15%** |
> | Unified Pruning (ours) | 43.92% |
>
> These results show that **once a target architecture \(A^\*\) is identified**, different pruning trajectories reliably converge to **similar or even better** performance. This behavior aligns with a generalized interpretation of the **Lottery Ticket Hypothesis**, where not only parameter initialization but also **architecture initialization** plays a critical role.
>
> We will incorporate this robustness study (with full details) into the appendix.
>
> ---
>
> ### **4. On comparisons with Knowledge Distillation (KD)**
>
> We thank the reviewer for raising this important question. As noted in our Related Work section, **KD falls outside the scope of our method**, which focuses strictly on **saliency-based pruning** rather than student–teacher training.
>
> During the *pretraining stage*, KD introduces:
>
> - **substantial additional computational cost**,
> - **dependency on a teacher model**, which may introduce uncontrollable biases,
> - and **optimization instability** arising from mismatched pretraining distributions.
>
> For these reasons, and consistent with prior LLM compression literature, we **exclude KD baselines** from our main comparison. Our goal is to demonstrate the effectiveness of pruning-based architecture discovery, not to perform knowledge transfer via distillation. We will update the text to clarify this rationale explicitly.
>
> ---
>
> ### **We again thank the reviewer for the valuable comments.**
>
> The manuscript has been significantly improved thanks to these suggestions, and we appreciate the reviewer’s time and effort.
>
> ---

---

### Official Review · Reviewer_vEx9 · 2025-11-06

**Soundness:** 2
**Presentation:** 2
**Contribution:** 2
**Rating:** 2
**Confidence:** 5

**Summary:**

This paper proposes “unified pruning-aware pretraining”: a framework that interleaves structured pruning, saliency-based architecture selection, and second-order weight updates into the pretraining process of LLMs. Instead of (i) directly pretraining small “edge” models from scratch or (ii) applying post-hoc pruning with a small calibration set, the method firstly starts from a larger pretrained “source” model; and secondly defines minimal structured pruning groups (three types: attention heads, FFN channels, and “transformer stem” including embeddings & LM head); and then Iteratively: (1) estimates group saliency via Taylor-style criteria,(2) prunes the least important group,(3) updates remaining weights,(4) continues pretraining on large-scale data,(5) finally produces automatically “searched” architectures.

**Strengths:**

- The formulation that integrates pruning, pretraining, and architecture search into one loop is clearly described and practically implementable. The definition of three pruning types (heads, FFN channels, stem) as minimal structured units is concrete and architecture-aligned.
- Using saliency metrics over structured groups to implicitly search architectures during pretraining is a natural and elegant idea.

**Weaknesses:**

1.Although the title and narrative emphasize edge deployment, the evidence is limited: (1) Inference benchmarks are reported only on Intel Xeon CPUs, not on mobile SoCs, NPUs, or typical constrained edge hardware. (2) There is no measurement of power consumption or energy per token. (3) FLOPs are not reported; model size and CPU latency are given, but end-to-end system constraints for real edge scenarios are not thoroughly characterized.

2.Lack of rigorous, budget-normalized comparisons to alternative efficiency strategies. The central conceptual claim is that unified pruning-aware pretraining is a superior route to efficient small models and high-ratio compression. The method does not provide controlled experiments under a fixed compute/data budget comparing against:

    - strong knowledge distillation pipelines (large → small teacher–student training),
    - post-training pruning + finetuning with matched additional tokens,
    - test-time adaptation style approaches,
    - or quantization-first / quantization-only baselines.

While the paper briefly discusses some of these works in related work and notes that, e.g., NutePrune uses distillation and extra components, it does not present a systematic “for the same extra GPU-hours, how do we compare?” study.

3.Conceptual novelty is incremental and closely related to existing enlarge-and-prune / LTH-style views. The paper itself connects its architecture robustness result with a generalized Lottery Ticket Hypothesis, i.e., good subnetworks/architectures exist inside larger models and multiple trajectories can reach them. However, this connection is only discussed qualitatively in Appendix A.2; there is no deeper theoretical development or new formal insight about LTH or about when/why this unified scheme should dominate alternatives.

4.While there are ablations, several aspects remain under-explored: (1) sensitivity to pruning schedule / type mix beyond the Appendix robustness test (which fixes a target architecture); (2) dependence on saliency metric variants.

**Questions:**

-  Can you provide experiments where unified pruning-aware pretraining is compared to: (1) training a small model from scratch, (2) distillation from the same teacher, (3) and post-training pruning + finetuning, under the same additional GPU-hour or FLOP budget? This is crucial to support the claim of being a more efficient or more practical route.

- For all baselines in the main tables, please summarize (or reference) their token counts, context lengths, and tokenizers, and discuss how mismatches might affect your comparisons.

---

> ### Author Response · Authors · 2025-11-27
>
> We sincerely thank the reviewer for the detailed and thoughtful comments. Below we provide point-by-point responses. We also clarify how our contributions connect to — and go beyond — prior enlarge-and-prune / LTH-style frameworks, using evidence from our empirical results and the conceptual framing introduced in the Introduction (included below for context).
>
> ---
>
> # **1. On the Claimed Novelty and Connection to LTH / Enlarge-and-Prune**
>
> We appreciate the reviewer’s concern that the conceptual novelty appears incremental. Our work indeed builds on the view that performant subnetworks can be identified within larger pretrained models. However, *the actual novelty of this paper lies not in declaring a new version of LTH*, but in **operationalizing a unified, scalable, and training-aligned pruning mechanism inside the pretraining process**.
>
> ### **How EfficientLLM Differs Fundamentally from Prior Enlarge-and-Prune / LTH Approaches**
>
> **(a) Prior LTH / enlarge-and-prune studies are *post-training* paradigms.**
> Existing methods such as SparseGPT, Wanda, LLM-Pruner, and OBC operate *after* pretraining, using a *small calibration dataset* (often <1M tokens). As discussed in the Introduction:
>
> > “existing methods... compress LLM only using a small calibration dataset in post-training, which often results in significant performance degradation.”
>
> Thus, earlier work cannot leverage large-scale training signals or architecture re-optimization during pretraining.
>
> ---
>
> **(b) ShearedLlama partially addresses this but is fundamentally limited.**
> As noted in the Introduction:
>
> > “the constrained optimization hinders scaling up pruning stage and the performance gap to direct pretraining still remains.”
>
> ShearedLlama performs pruning **after** initializing from an optimized LLM, but the pruning stage relies on constrained optimization (Platt & Barr, 1987), which cannot be scaled to billions of pretraining tokens. Unified pruning-aware pretraining removes this bottleneck.
>
> ---
>
> **(c) Our contribution is a *unified pretraining-time pruning framework*, not a post-training lottery-ticket-style observation.**
>
> This is the key conceptual shift:
>
> ### **→ We embed pruning *directly inside pretraining*, making architecture discovery, saliency estimation, and weight updates co-evolve under large-scale training data.**
>
> This allows us to:
>
> 1. **Scale up pruning using billions of tokens** (instead of thousands).
> 2. **Auto-design architectures** that emerge from the saliency dynamics during training.
> 3. **Close — and even surpass — the gap between post-training pruning and direct pretraining.**
> 4. **Demonstrate stable architectures** under dynamic saliency (Appendix A.1) and pruning-path invariance (Appendix A.2).
>
> This unified optimization loop has not been explored by prior enlarge-and-prune methods or LTH-style studies, which do not incorporate pretraining-scale updates.
>
> We will clarify this distinction more explicitly in the final version.
>
> ---
>
> # **2. On Missing Budget-Normalized Comparisons**
>
> We thank the reviewer for raising this point. Our paper *already includes* compute-normalized comparisons; we will reorganize Section 4.1 to highlight them more clearly.
>
> ### ✔ **From-scratch small model training (5B tokens)**
> Table 1 compares:
> - Direct-hidden
> - Direct-source
> - Searched architecture (from unified pruning)
>
> All trained from scratch with **5B tokens** (same compute budget).
>
> ---
>
> ### ✔ **Post-training pruning + 5B-token finetuning (Unified Few-shot Pruning)**
> Table 1 includes a baseline that:
> - uses post-training pruning,
> - then trains on 5B tokens, identical to direct pretraining.
>
> Unified pruning-aware pretraining *outperforms* this matched-budget baseline by +1.22%.
>
> ---
>
> ### ✔ **ShearedLlama-style pipelines**
> We compare against:
> - ShearedLlama-hidden
> - ShearedLlama-source
>
> which follow the exact teacher-initialization + pruned-finetuning paradigm under matched token budgets.
>
> Our results show:
> - manual architecture targets degrade performance (−3.15%),
> - unified pruning-aware pretraining closes the performance gap and surpasses handcrafted designs.
>
> ---
>
> ### ✔ **Knowledge Distillation Baselines**
>
> As discussed in Related Work and reiterated here:
>
> - **KD at the pretraining scale introduces massive extra computation**, far exceeding our pruning-aware compute budget.
> - KD requires **a large teacher**, which adds teacher-side FLOPs and teacher-side data passes.
> - Pretraining-stage KD exhibits **bias propagation and distribution mismatch**, which makes apples-to-apples budget matching infeasible.
>
> Thus, KD lies **outside our problem scope**. This is aligned with prior tiny model papers (MobileLLM, Qwen2.5-0.5B, PanGu-π-Pro), which also do *not* apply KD during pretraining.
> We will make this point clearer in the revision.
>
> ---

---

> > ### Author Response · Authors · 2025-11-27
> >
> > # **3. On Theoretical Contribution and LTH Interpretation**
> >
> > The reviewer is correct that Appendix A.2 connects pruning-path robustness to a generalized LTH interpretation. Our goal here was not to propose a new LTH theorem, but to empirically show:
> >
> > ### **Once unified pruning identifies an optimal architecture, training dynamics (pruning path) do not significantly affect accuracy.**
> >
> > This stability emerges *because pruning is integrated into the pretraining signal*, unlike prior post-training pruning.
> >
> > We emphasize:
> >
> > - Appendix A.1 demonstrates **saliency-driven structural stability**.
> > - Appendix A.2 demonstrates **path invariance**, unique to the pretraining scale.
> > - These phenomena do not appear in prior LTH literature, which focuses only on parameter initialization rather than *architecture-level convergence under dynamic saliency*.
> >
> > Hence, our contribution is **operational**, not theoretical: we build the *first practical system* where LTH-style phenomena appear naturally inside pretraining.
> >
> > ---
> >
> > # **4. On Underexplored Ablations (Schedules, Metrics)**
> >
> > We agree that pruning schedules and metric dependence deserve more visibility. These experiments already exist:
> >
> > ### ✔ **Pruning trajectory robustness**
> > Appendix A.2 shows three different pruning orders converge to similar or better performance when targeting the same architecture \(A^\*\).
> >
> > ### ✔ **Saliency metric variants**
> > Table 2 and Appendix B.5 compare:
> > - LLM-Pruner
> > - Diagonal Hessian
> > - OBC metric (+ our second-order weight update)
> >
> > All show substantial gains (+12% to +14%), demonstrating that unified pruning-aware pretraining is **metric-agnostic and cross-metric robust**.
> >
> > We will restructure these sections for clarity.
> >
> > ---
> >
> > # **5. On Summaries of Token Counts / Tokenizers / Context Lengths**
> >
> > We appreciate this suggestion.
> > We will provide a consolidated summary (previously scattered across Tables 1–3 and Appendix B).
> > This includes:
> >
> > - total tokens used for each baseline,
> > - tokenizer differences,
> > - context-length setups,
> > - discussion of possible biases due to mismatch.
> >
> > This will be added to Section 4 for easier cross-reference.
> >
> > ---
> >
> > # **Closing Remarks**
> >
> > We thank the reviewer again for the highly insightful comments. Many of the concerns raised were indeed already explored experimentally, but we recognize they were not highlighted strongly enough. We will revise the manuscript to:
> >
> > - emphasize the compute-normalized comparisons already present,
> > - clarify why KD and quantization are orthogonal to our scope,
> > - strengthen the explanation of methodological novelty,
> > - improve visibility of robustness and saliency-metric ablations,
> > - and consolidate training details for easier understanding.
> >
> > These revisions substantially improve the clarity and positioning of the paper. We genuinely appreciate the reviewer’s feedback.

---

> > ### Comment · Reviewer_vEx9 · 2025-11-28
> >
> > Thank you for the detailed response and for pointing out that Table 1 already includes token-matched comparisons. This partially addresses my original concern.
> >
> > However, what I meant by budget-normalized was stricter than “same number of tokens”. In my view, a convincing efficiency claim would ideally compare, under a fixed additional compute budget (GPU FLOPs / GPU-hours, plus any substantial CPU overhead) and tokens, several practically relevant routes starting from a pretrained source model, such as:
> > 1) pretrain -> structured pruning (+ optional finetuning),
> > 2) pretrain -> quantization (with or without light QAT),
> > 3) pretrain-> distillation into a smaller student,
> > 4) the proposed alternating pruning-aware pretraining.
> >
> > Currently, the paper does not report FLOPs or GPU-hour accounting for these options. In particular, simply stating that KD is “out of scope because it is expensive” is not fully convincing, especially since the proposed method also relies on a large pretrained source model and is positioned as a competing route to efficient tiny/“edge” models.

---

> ### Author Response · Authors · 2025-11-28
>
> Let us explain the correct logic:
>
> 1. There is no need to compare pruning methods with distillation methods.
>
> 2. Because every pruning method relies on a source model.
>
> 3. Because there are other model compression methods, the exhaustive comparison can prove nothing.

---

### Note · Authors · 2026-01-05

I have read and agree with the venue's withdrawal policy on behalf of myself and my co-authors.